# Defining the Relationship of Gut Microbiota, Immunity, and Cognition in Early Life—A Narrative Review

**DOI:** 10.3390/nu15122642

**Published:** 2023-06-06

**Authors:** Melissa Stephanie Kartjito, Mikhael Yosia, Erika Wasito, Garry Soloan, Achmad Furqan Agussalim, Ray Wagiu Basrowi

**Affiliations:** 1Medical and Science Affairs Division, Danone Specialized Nutrition Indonesia, Jakarta 12950, Indonesia; melissa.kartjito@danone.com (M.S.K.); erika.wasito@danone.com (E.W.); ray.basrowi@gmail.com (R.W.B.); 2Faculty of Medicine, Universitas Indonesia, Jakarta 10430, Indonesia; garry@ui.ac.id (G.S.); achmad.furqan91@ui.ac.id (A.F.A.)

**Keywords:** gut microbiota, immunity, cognition

## Abstract

Recently, the immune system has been identified as one of the possible main bridges which connect the gut–brain axis. This review aims to examine available evidence on the microbiota–immunity–cognitive relationship and its possible effects on human health early in life. This review was assembled by compiling and analyzing various literature and publications that document the gut microbiota–immune system–cognition interaction and its implications in the pediatric population. This review shows that the gut microbiota is a pivotal component of gut physiology, with its development being influenced by a variety of factors and, in return, supports the development of overall health. Findings from current research focus on the complex relationship between the central nervous system, gut (along with gut microbiota), and immune cells, highlighting the importance of maintaining a balanced interaction among these systems for preserving homeostasis, and demonstrating the influence of gut microbes on neurogenesis, myelin formation, the potential for dysbiosis, and alterations in immune and cognitive functions. While limited, evidence shows how gut microbiota affects innate and adaptive immunity as well as cognition (through HPA axis, metabolites, vagal nerve, neurotransmitter, and myelination).

## 1. Introduction

The first 1000 days of life are crucial moments in human development that provide a unique opportunity to shape lifelong health conditions. During this early period, the establishment of gut microbiota is influenced by a complex interaction between maternal health, nutrition, metabolic status, and various other internal and external factors. It is widely believed that the maturation of the gut microbiota is completed before the third year of life [1]. Through recent understanding of the gut–brain axis, this period of rapid gut microbiota establishment would be crucial to the development of the brain and subsequent cognitive function.

Recently, the immune system has been identified as one of the possible main bridges which connect the gut–brain axis [2]. The association between intestinal inflammatory disorder and behavioral and neuropsychological problems has been established to a certain extent. Conversely, issues in the spine and brain have also manifested with gastrointestinal complications [3]. Studies have tried to explain such a phenomenon through several theories, one of them being that the leakage of microbiota through a breach in gut integrity may result in chronic inflammation that would release systemic inflammatory cytokines, affecting the central nervous system (CNS) causing changes in behavior, mood, and stress level [4]. Analysis has also revealed that certain microbes, such as *Enterobacteriaceae*, are more likely to induce a systemic inflammatory response. On the other hand, bacteria such as *Lactobacillus* secrete acid, which can create an environment in the gut that is less favorable for pro-inflammatory microbes, leading to a somewhat anti-inflammatory condition [5].

The immune system and nervous system are also the two primary regulators of homeostasis in the body, communicating and relying on each other to ensure a normal functioning organism. Components of the immune system, particularly microglia, play a crucial role in the development and activity of the nervous system; the microglia have been known to constantly monitor synapses in the parenchyma of the central nervous system and affect its development early in life [6]. Therefore, a properly functioning immune system in a child is critical for the development of cognitive functions and neurogenesis.

Considering the intricate relationship between gut microbiota, immunity, and cognition, it can be assumed that disruption in the gut microbiota may lead to changes in the immune system’s effectiveness that will cause impairment in the central nervous system and cognitive functions. Conversely, maintaining and supporting a healthy microbiota may contribute to the proper development of the immune system and cognitive functions. This assumption, however obvious, still needs to be analyzed further. This review examines the available evidence on the microbiota–immunity–cognitive relationship and its potential effects on early human health, specifically focusing on the critical period of gut microbiota–immunity–cognitive development during the first three years of life, in which the gut may influence immunity and cognitive development.

## 2. Materials and Methods

### Literature Search

The objective of this review was to analyze and synthesize available literature to understand the relationship between gut microbiota, the immune system, and cognition in the pediatric population. A comprehensive literature search was conducted in March 2022 across databases including Google Scholar, PubMed, EMBASE, and Cochrane. The search strategy included the terms “gut,” “microbiota,” “immunity,” and “cognitive.” Additionally, a MeSH term, “((Gut[All Fields] AND Microbe[All Fields]) AND (“immunity” [MeSH Terms] OR Immunity[Text Word])) OR Cognitive[All Fields],” was employed to refine the search across the databases.

Each article was individually assessed for its relevance through the screening of titles and abstracts. Duplicate entries were removed, and studies that did not explore the interaction between gut microbiota, immunity, and cognition in the pediatric population were excluded. The full texts of the remaining articles were carefully examined, and relevant data were extracted for further analysis.

To ensure the quality of the included studies, each was evaluated based on its methodology, sample size, statistical analyses, and the relevance and significance of its findings. Studies that lacked significant or sufficient information regarding the relationship between gut microbiota, immunity, and cognition, or had methodological limitations were excluded from the review.

The results of the review are presented systematically. The discussion and conclusion drawn from the review are based on the synthesized findings of the studies, providing a comprehensive understanding of the interaction between gut microbiota, the immune system, and cognition in the pediatric population.

## 3. Microbiota and Immunity—Consequences of Gut Colonization

Microbiota of the gut heavily affects the development and maturation of the immune system, especially in the development of tolerance towards ingested antigens in the gut. The tolerance is developed to ensure that non-harmful antigens do not trigger an inflammatory response in the gut—with the suggestion that failure in developing tolerance will result in inflammatory-related diseases in the gut later in life.

### 3.1. Effects on Innate Immunity Development

The intestinal mucosa, along with the membrane, acts as the first line of defense against pathogens that invade the gastrointestinal tract. This barrier is made up of a thick extracellular layer and a thinner yet complex inner layer. The inner layer is made up of intestinal epithelial cells (IECs), goblet cells, and various membrane-bound mucins and glycolipids attached to IECs. The extracellular layer is made up of mucus that contains antimicrobial peptides (AMPs), secreted IgA (sIgA), and secreted mucin. The secreted mucin provides a principal energy source for gut microbiota, as well as a protective barrier toward pathogen invasion [7]. AMPs are proteins that confer protective effects against pathogenic bacteria, certain fungal species, protozoa, and viruses. Well-known AMPs that play a major role in gut immune defense are the α- and β-defensin produced by Paneth cells [8]. Meanwhile, sIgA antibodies modulate the microbial colonization of the epithelium and prevent the adsorption of pathogenic microbes [9].

It is evident from current studies that gut microbiota significantly influences the development and homeostasis of barrier components. For instance, the production of secreted mucins is modulated by gut microbiota. Factors such as the presence of pathogenic bacteria and poor diet can disrupt the process of mucin glycosylation, which is crucial for physiological protection and cellular communication, including signal transduction and cell-to-cell adhesion [10,11]. The absence of physiological commensal gut microbes and the presence of pathogenic species such as *B. hyodysenteriae* and *Helicobacter* suis can alter the normal glycosylation process, contributing to the development of inflammatory diseases such as IBD, Crohn’s disease, and colorectal cancer [12,13,14].

Certain microbes are also known to support the production of antimicrobial peptides (AMPs). Vaishnava et al. demonstrated that Paneth cells can detect enteric bacteria via MyD88 signaling, a crucial step for bacterial-induced secretion of AMPs, thus protecting against the penetration of pathogenic bacteria such as *Salmonella* sp. [15].

In addition to the microbes themselves, various biologically active metabolites secreted by gut microbiota are known to modulate immunity and maintain immune homeostasis. Short-chain fatty acids (SCFAs) such as acetate, butyrate, and propionate, which account for 90–95% of all colonic SCFAs, are produced by the fermentation of dietary fiber by commensal microbiota [16]. These SCFAs regulate adaptive immunity by binding to various G protein-coupled receptors (GPCRs). The three SCFAs can interact with GPR43, a receptor that recognizes a wide range of SCFAs. The interaction of acetate with the GPR43 pathway positively influences intestinal IgA response, with retinoic acid, a metabolite of vitamin A, acting as the mediator [17,18].

Furthermore, SCFAs are known to regulate the colonic regulatory T cell (Treg) pool by inhibiting histone deacetylase, subsequently increasing the acetylation and expression of the Foxp3 gene, a key transcription factor promoting Treg differentiation [19,20]. Beyond modulating IgA response and Treg differentiation, SCFAs also nourish and promote the proliferation of CD4+ T cells and CD8+ T cells, underscoring the importance of gut microbiota in the systemic response against invading bacterial or viral pathogens. Kespohl et al. demonstrated the low-dose effect of butyrate in amplifying Foxp3 transcription factor and subsequent Treg differentiation, while concurrently producing pro-inflammatory molecules that aid the function of conventional T cells. [19] SCFAs also enhance the functionality of memory CD8+ T cells, presumably through butyrate, which promotes oxidative phosphorylation and fatty acid catabolism as the principal method for CD8+ T cell metabolism [21].

### 3.2. Effects on Adaptive Immunity Development

The adaptive immune mechanisms serve as a secondary line of defense against invasive pathogens, notable for their specific responses towards these pathogens. Within the gastrointestinal (GI) system, both bacterial components and active metabolites contribute to the development and maintenance of adaptive immune system homeostasis. Certain microbes, such as Bacteroides fragilis and Bacteroides thetaiotaomicron, can penetrate the mucus layer, colonize intestinal epithelial cells (IECs) and colonic crypts, and interact with *Mucispirillum schaedleri*, which resides in the cecal crypt. These microbes are presented to the immune system’s dendritic cells, allowing a minor population of these commensal microbes to enter and localize within the mesenteric lymph node, thereby stimulating an effective mucosal immune response [22,23,24,25,26,27].

Adaptive humoral immunity within the GI tract primarily operates through secretory Immunoglobulin A (sIgA) antibodies, which are responsive to the commensal gut microbiota. An earlier section of this paper examined the critical role gut microbiota play in establishing and maintaining the function of sIgA as a component of the intestinal immune barrier. The presence of intestinal plasma cells facilitates the production of sIgA through either T cell-independent or T cell-dependent mechanisms. Notably, most commensal gut microbiota trigger sIgA responses via T cell-independent pathways [21]. Certain unusual commensal bacteria, such as *Mucispirillum*, interact with antigen-presenting cells to incite adaptive T cell and B cell responses for sIgA production, as these particular microbes lack sites for T cell-independent antigen-induced sIgA [28]. Flagellated commensal bacteria also promote sIgA production, as the protein flagellin can activate Toll-like receptor (TLR) 5 on dendritic cells, which subsequently stimulates naïve B cells to differentiate into plasma cells that produce IgA [29].

Finally, the production of sIgA is influenced by intestinal microorganisms. Gut microbiota affect sIgA production by expressing specific microbe-associated molecular patterns (MAMPs) that activate the polymeric immunoglobulin receptor (pIgR). This leads to a process that enables the transportation of dimeric IgA from plasma cells within the lamina propria, through IECs, to eventually display the sIgA on the apical surface of the intestinal mucosa [30,31]. Several in vitro studies have shown that certain microbes, such as Bacteroides thetaiotaomicron, stimulate increased pIgR expression, while *E. coli* is known to produce a highly potent, long-lived sIgA response in germ-free mice colonized with non-dividing *E. coli*. [32,33]. The sIgA response is adaptive to the current state of gut microbiota. For instance, introducing new microbes to mice previously colonized with *E. coli* results in a decrease in the sIgA response to the original *E. coli* colonizer [31].

## 4. Microbiota and Cognition

In addition to the importance of gut microbiota, the first 1000 days also represent a critical period for neurocognitive development. An animal study led by Braniste and colleagues revealed that germ-free mice demonstrated disruptions in the blood-brain barrier and myelin formation, underscoring the potential role of the microbiome in cognitive development [34]. Further animal studies corroborate these findings, showing that when gut microbes from autism spectrum disorder (ASD) patients are introduced to mice, these animals begin to exhibit ASD-like behavior and an early onset of anxiety (occurring at 3 weeks instead of the usual 10) [35,36]. Similar findings have been replicated in human studies; for example, children with ASD exhibit changes in their gut microbiome, which appear to influence cognitive behavioral development as they grow older [36,37]. A recent review by Sittipo et al. also emphasized the critical role gut microbiota plays in modulating brain development and function, particularly through the microbiota–gut–brain axis, and it is implicated in the regulation of central nervous system (CNS) function via development and maturation of the immune and endocrine systems [38]. Additionally, this research underscores the influence of gut microbiota on the integrity of the blood-brain barrier, suggesting its potential role in neuroimmunity and prevention of neuroinflammation [38].

Though the area is not yet fully explored, alterations in the microbiota have been found to affect specific brain regions, including the hippocampus, the hypothalamic–pituitary axis (HPA), and structures related to fear response. Gareau and colleagues demonstrated that germ-free mice, when infected with *Citrobacter rodentium*, exhibited impairments in non-spatial recognition and working memory associated with the hippocampus upon exposure to acute stress. This was evidenced by reductions in brain-derived neurotrophic factor (BNDF) and c-fos, proteins vital for hippocampal memory regulation [39,40,41]. The hypothalamic–pituitary axis, a brain region responsible for homeostatic regulation, is also affected. With regards to fear, in a pilot study by Carlson et al., a one-year-old infant with high levels of *Veillonella, Dialister*, *Clostridiales,* and low levels of *Bacteroides* showed a negative association with both the prefrontal cortex and the amygdala, which are brain structures regulating fear behavior [42,43].

The gut microbiome communicates with the brain through various multidirectional pathways that also involve other systems. Some of the key pathways are listed below:Hypothalamic–Pituitary–Adrenal (HPA) axis: A study by Gareau in 2014 discovered that germ-free mice exhibited an increase in HPA hyperactivity. Problems with the HPA axis in human gut dysbiosis can result in behaviors indicative of anxiety and depression [41].Metabolites: Microbes in the gut produce metabolites, including toxins and short-chain fatty acids (SCFAs), that can induce an inflammatory response. This inflammation can alter the blood-brain barrier, facilitating the transport of molecules [42,44].Vagus Nerve: Short-chain fatty acids (SCFAs) can stimulate the vagus nerve, triggering the release of neurotransmitters necessary for the gastrointestinal tract [42,44].Neurotransmitters: The gut microbiome can modulate certain neurotransmitters, as seen with dopamine and serotonin production. Decreased levels of these neurotransmitters can directly affect a patient’s overall mood [45].Myelination: Gut microbiota can induce alterations in myelination, which can affect various brain regions, particularly the prefrontal cortex, a region responsible for decision making. In a study conducted by Lu and colleagues, germ-free mice displayed decreased myelination [42,46].

## 5. Immunity and Cognition

Recent research has highlighted the role of immune system activity in child cognitive development, driven by evidence linking infectious diseases and neonatal stress to long-term impacts such as altered immunological reactivity and behavioral disorders [47]. Contemporary literature has examined the correlation between postnatal immune system activity and cognitive functions from various perspectives. For instance, maternal inflammation during pregnancy, antenatal infection in preterm neonates, and neonatal infection have all been suggested as risk factors for impaired IQ development and neuropsychiatric disorders, such as depression and schizophrenia [48].

Miller and colleagues conducted a genome-wide transcriptional profiling study among healthy adults with varying socioeconomic status (SES) levels. They found that individuals who had experienced low SES in early life exhibited significant upregulation of genes from the CREB/ATF transcription factor family, which plays a crucial role in transmitting adrenergic signals to leukocytes. Additionally, these individuals showed significant downregulation of genes coding for glucocorticoid receptor response elements. This data suggests that early-life psychosocial stress may lead to altered gene expression, resulting in a defensive phenotype characterized by poor glucocorticoid signaling and heightened adrenocortical and inflammatory responses [49].

One particularly well-studied example of early life immune compromise and its long-term cognitive consequences can be seen in the association between viral infection and autism spectrum disorder (ASD). Epidemiological data suggest that exposure to viruses such as rubella may increase the risk of autism [50,51]. While the exact pathophysiological mechanisms behind viral infections and subsequent ASD remain unclear, emerging evidence indicates that neonatal infection can induce a pro-inflammatory state. This is evidenced by consistently observed alterations in cytokine profiles among children with ASD [52]. Elevated levels of pro-inflammatory cytokines such as IL-6, IL-23, IL-12, and TNF-α have been noted in 29 children diagnosed using the Childhood Autism Rating Scale (CARS), suggesting a dysregulated immune response [53]. This chronic cytokine activation leads to immune consequences. For instance, production of the pro-inflammatory macrophage chemoattractant protein (MCP)-1 and TGF-β1 can significantly activate astroglia and microglia, as evidenced by immunocytochemical studies [54].

In vitro model studies further illuminate how cytokines might modulate the pathogenesis of ASD, showing, for example, that elevated IL-6 in the brains of rat models can impair synapse formation, neuronal circuit balance, and dendritic spine formation [55].

Such evidence has led to the understanding that exposure to adverse life events during the first years of life can increase the risk for a wide range of behavioral, neural, and psychological disorders. This challenges the idea that the critical period for neurodevelopment is restricted to the first 1000 days of life [56]. Instead, it suggests a complex interplay between host susceptibility and environmental factors, spanning from conception through early and middle childhood (3–12 years old). Infections such as intestinal helminths, malaria, and schistosomiasis among school-aged children have all been associated with cognitive impairment, even in the absence of direct central nervous system (CNS) effects. However, it is likely that these processes involve inflammation, and the burden of such infections may even outweigh those that directly affect the CNS, such as viral encephalitis, bacterial meningitis, and HIV infection [57].

## 6. Discussion

The human gut is host to a diverse array of microorganisms, with the colonization process beginning at birth and continuing through early childhood. This process is influenced by various factors such as mode of delivery, breastfeeding and the introduction of solid food, and gestational age, among others. The gut microbiota plays critical roles including metabolic, protective, and trophic functions, such as nutrient breakdown, pathogen prevention, and immune system homeostasis [58,59]. Recent stratification methods, such as the concept of enterotypes, help in understanding gut microbiota and their functions, however, there can be variations in their grouping and interpretation [60].

The first few years of life are pivotal for microbiome development, with various factors, including mode of birth, antibiotic use, and dietary choices, influencing the microbiome diversity (Figure 1). Vaginal delivery, for example, exposes infants to a diverse microbiota, while cesarean section births can lead to delayed colonization of certain beneficial bacteria [61,62,63,64,65,66,67]. Antibiotic use can disrupt gut microbiota, potentially leading to dysbiosis and increased susceptibility to infection [68,69,70,71,72,73,74]. Early feeding practices and the introduction of solid food also play a significant role in shaping the gut microbiota, with breastfeeding supporting the growth of beneficial microbes including *Bifidobacterium* [75,76,77,78,79,80,81,82,83,84], and the introduction of solid foods influencing the diversity and abundance of certain bacterial families [85,86,87,88,89]. Finally, the gestational age can affect the development of gut microbiota, with preterm infants often having a lower diversity of gut microbiota and a higher risk of immature immune response [90,91,92,93,94].

Advancements in the field of sequencing technologies and bioinformatics tools allow identification of the variety of microbial species that exist within the gut. Clinically, it is often relevant to categorize these microbes as “good” and “bad” microbes that positively impact health status or contribute to the occurrence of diseases, respectively. Some examples of well-studied “good bacteria” include, *Bifidobacterium* sp., and *Lactobacillus* sp. On the other hand, several well-studied “bad bacteria” include *Enterococcus faecalis, Clostridium difficile*, and *Campylobacter* sp. [5].

The gut microbiota and the immune system share a bidirectional relationship that contributes to maintaining a healthy gastrointestinal (GI) system and preventing disease onset. Interactions between immune and epithelial cells with the metabolites and nutrients produced by microbes drive the development of both innate and adaptive immune systems [95,96].

The gut’s colonization process begins at birth, when a baby passing through the vagina ingests a bolus of microbes, which then proliferate in the GI system. This microbial diversity expands with the introduction of breast milk and supplementary foods as the infant grows [96,97]. The innate immunity is characterized by an intra- and extracellular layer in the intestinal lining, composed of epithelial cells, mucins, antimicrobial peptides (AMPs), and secretory Immunoglobulin A (sIgA), functioning to prevent pathogen entry. Additionally, variations in immune cells, including antigen-presenting cells, neutrophils, mast cells, and natural killer cells, are observed between germ-free and microbe-populated mice, showcasing their role in pathogen defense and tolerance towards normal gut microbiota [98,99,100]. These interactions between microbes and intestinal epithelial cells may elicit either a protective or a tolerance response.

Adaptive immunity serves as a secondary defense mechanism, characterized by its distinct response to pathogens. sIgA antibodies, for instance, prevent pathogens from adhering to intestinal epithelial cells [96]. B and T cells also contribute to the adaptive immune system’s regulation. Molloy’s 2012 study highlighted the critical role of T regulatory cells, specifically FoxP3+ Treg cells, in establishing a tolerant response towards gut microbes and in suppressing the immune effects of dendritic cells (DC), T cells, and B cells in the intestine. Consequently, mutations in these cells could potentially lead to autoimmune disorders [97].

The immune system contributes to neurodevelopment through various mechanisms, one of which involves neuropeptides generated in the brain that modulate immune cells throughout the body. Receptors on both leukocytes and lymphocytes specifically interact with certain neuropeptides such as Adrenocorticotropic Hormone (ACTH), opioids, and enkephalins, facilitating their role as signaling molecules between immune cells [101,102]. Studies have highlighted enkephalins’ capacity to suppress immune responses, while leukocytes bound with opioids have been observed to decrease pain responses upon the release of corticotrophin-releasing hormone and norepinephrine, which typically increase during inflammation [102,103].

Microglia, the brain’s principal immune cells, influence brain development in myriad ways. Given their ability to produce growth and noxious factors, microglia are essential in maintaining the brain’s cellular population, contributing to cell death via the production of reactive oxygen species and nerve growth factors. They also clear both living and dead cell bodies during brain development to facilitate a growth-conducive tissue environment. Simultaneously, microglia promote synaptogenesis, synaptic pruning, and angiogenesis through various, unspecified cytokines [104,105,106]. However, increased immune cell activity during neurodevelopmental periods could result in aberrant microglial development. This aberration could lead to reduced synaptic pruning, lower neurotransmitter levels, and poor connectivity, potentially culminating in disorders like schizophrenia and autism-like behavior [107,108].

Upon infection or trauma, various brain cells, with microglia being the major contributor, produce cytokines. When cells become infected or damaged, they release pathogen-associated molecular patterns (PAMPs) and danger-associated molecular patterns (DAMPs), instigating the production of pro-inflammatory cytokines that attract immune cells to the infection site [109]. Multiple studies also indicate that cytokines facilitate processes such as neurogenesis and differentiation in the brain. Table 1 details some cytokines and their roles in neurodevelopment.

### 6.1. Possible Gut-Immune-Cognition Interactions

The current focus of research is shifting towards the intricate tripartite relationship between the central nervous system (CNS), gut, and brain immune cells, along with gut microbiota. Maintaining a balanced interaction among these systems is crucial for preserving homeostasis and preventing complications within the gut–brain and gut–immune axes. Within the CNS, immune regulation is governed by the blood-brain barrier, in conjunction with microglia, astrocytes, and oligodendrocytes. Microglia are macrophages that help to maintain brain homeostasis and can be activated by various microbial and immune pathways [115,116]. Bercik et al.’s animal study showed the presence of microglia in germ-free mice; however, these cells displayed characteristics of immature microglia with compromised function. This finding aligns with those from Schafer’s study, which highlighted abnormal microglia development in germ-free mice lacking *Bacteroides distasonis* and *Lactobacillus salivarius* [6,117]. Short chain fatty acids (SCFAs), a byproduct of gut microbiota, can also instigate microglial activation via G protein receptors, thus triggering an inflammatory response [118].

Astrocytes, another group of brain support cells, also participate in immune and microbiota activation. They carry aryl hydrocarbon receptors (AHRs) capable of binding to gut microbiota metabolites, thereby eliciting an anti-inflammatory response. The metabolite tryptophan can activate both microglia and astrocytes. According to Gareau et al., mice with experimental autoimmune encephalomyelitis (EAE) showed reduced symptoms when treated with tryptophan [40,115].

Interactions between the gut microbiota, immune system, and brain can be triggered during infection, neurogenesis, or medical treatments. For example, segmented filamentous bacteria (SFB) induce EAE symptoms in germ-free mice by activating Th17 cells, which infiltrate the CNS. Conversely, colonization of *B. fragilis* in germ-free mice leads to the production of IL-10 specific T regulatory cells [119,120]. Several studies have investigated the influence of gut microbes on neurogenesis and myelin formation, suggesting that their beneficial effects are time-sensitive and most prominent during early infancy [121]. In autism spectrum disorder, symptom exacerbation has been linked to gut dysbiosis. However, symptoms can be partially alleviated by the administration of probiotics including *B. infantilis* and *L. reuteri* [122,123].

As cancer incidence surges globally, chemotherapy has become a prevalent treatment approach. However, it is important to be aware of potential long-term complications arising from this modality. Prominent long-term effects of chemotherapy include dysbiosis and alterations in immune and cognitive functions, with notable conditions encompassing decreased decision-making capability, slower processing speed, and reduced dexterity [124,125,126]. Notably, 22.4% of the 90.5% of patients undergoing chemotherapy were found to experience depression and other psychosocial effects compared to cancer patients receiving other treatment types. Furthermore, chemotherapy-induced dysbiosis is ubiquitous across all age groups. The chemotherapy driven dysbiosis of the microbiota–gut–brain axis model illustrates how chemotherapy triggers dysbiosis and increased intestinal permeability (leaky gut), thus allowing toxins to enter the gastrointestinal system, stimulating pro-inflammatory cytokine production, and leading to systemic inflammation. This inflammatory state is exacerbated by cortisol release, which enhances the inflammatory process. In response to dysbiosis and inflammation, the hypothalamic–pituitary–adrenal (HPA) axis decreases serotonin production, potentially leading to cognitive and psychological dysfunction in patients [124,125]. Fatigue and weakness in patients might be attributed to the decrease in *Firmicutes* and *Actinobacteria*, and the increase in *Proteobacteria*, causing energy and vitamin imbalances and increasing the risk of mucositis [124,126,127]. The onset of “leaky gut” can trigger an inflammatory response, resulting in symptoms such as lethargy, depression, anorexia, and social withdrawal through the modulation of pro-inflammatory cytokines (TNF-a, IL-6, IL-1B)^142. These proposed relationships are summarized in Figure 2.

### 6.2. Future Implications

This review underscores the significance of a healthy gut microbiota in maintaining gut homeostasis, and how its perturbation may serve as a cause or risk factor for the emergence of certain gastrointestinal pathologies. It is hoped that these insights will elevate awareness regarding gut microbiota and their crucial role in overall health, given the significant burden associated with gastrointestinal diseases. It is widely acknowledged that the process of gut colonization in infancy forms a cornerstone of good health, given its implications for short-term and long-term health outcomes. For instance, infants with less diverse gut microbiota and elevated levels of potentially pathogenic microbes face a higher risk of necrotizing enterocolitis (NEC) [129]. Moreover, the establishment of early-life microbiota exerts long-lasting impacts, primarily through sustained interactions with the immune system. Consequently, abnormal gut microbiota development during this critical period is linked to the onset of allergy-related conditions including asthma and atopic eczema, and certain metabolic disorders (e.g., diabetes or obesity) [130,131]. Bacteria such as *Staphyloccocus* and *S. aureus*, for example, can meddle with peripheral metabolism, potentially triggering obesogenic effects [132].

The ramifications of these gastrointestinal disturbances may extend beyond health outcomes. In irritable bowel syndrome (IBS), where gut microbiota plays a significant role in its pathogenesis, a notable decline in quality of life is observed alongside an escalating socioeconomic burden globally. For instance, in the USA, IBS patients reportedly experience up to 9 days of reduced productivity monthly, resulting in a staggering economic loss of up to USD 20 billion annually due to presenteeism, absenteeism, and diminished daily productivity [133]. IBS is associated with 3.6 million physician visits yearly and a 50% surge in healthcare costs compared to non-IBS individuals [134,135].

On the other hand, there needs to be more emphasis on dietary interventions such as prebiotic, synbiotic, probiotics, and polyunsaturated fatty acids (PUFAs), which can be beneficial in mitigating the impact of early life insults on gut microbiota. Probiotics, which are live beneficial bacteria, can help in establishing a healthy gut microbiota, promoting gut integrity, and boosting immune function. PUFAs, especially omega-3 fatty acids, have anti-inflammatory properties and have been shown to influence the composition of the gut microbiota. Both probiotics and PUFAs can potentially help restore a healthy balance of gut microbiota after disruptions, thus enhancing gut health and immune function.

Despite the absence of studies that directly or indirectly quantify the cost of gastrointestinal pathologies associated with gut microbiota disturbances in Indonesia, there are some studies illustrating a correlation between low socioeconomic status and the prevalence of potentially pathogenic bacteria among Indonesian children [136]. Additionally, Surono et al.’s study comparing the microbiota profiles between stunted children and those with normal nutritional levels found significantly lower levels of certain microbes, such as *Prevotella 9*, among stunted children. This finding also highlights the subpar dietary intake among these children [137]. These insights hint at the potential benefits of implementing a nationwide program to supplement nutrient sources including synbiotics, prebiotics, probiotics, and fiber. Such initiatives could nourish gut microbiota development and subsequently inhibit the progression of pathological conditions.

## 7. Conclusions

Gut microbiota is a key element of gut physiology that not only shapes overall health but is also influenced by various factors. This review has highlighted gut microbiota and their significant implications for human health, especially regarding the gastrointestinal, immune, and nervous systems. The gut microbiota, immune system, and central nervous system are interconnected through a complex “gut-brain axis” that communicates via neural, endocrine, immune, metabolic, microbial, and barrier mechanisms, with disruptions potentially leading to various health issues. It underlines the need for increased awareness of gut microbiota’s role in health, particularly given the observed impacts on children. To gain deeper insights, the authors advocate for further primary research involving a larger sample size specifically within the pediatric population. This could help elucidate precise factors shaping gut microbiota development and the consequences of its disruption. Additionally, exploring the potential for a nationwide nutrition supplementation program to foster healthy gut microbiota could provide valuable solutions for current pediatric nutritional challenges.

## Figures and Tables

**Figure 1 nutrients-15-02642-f001:**
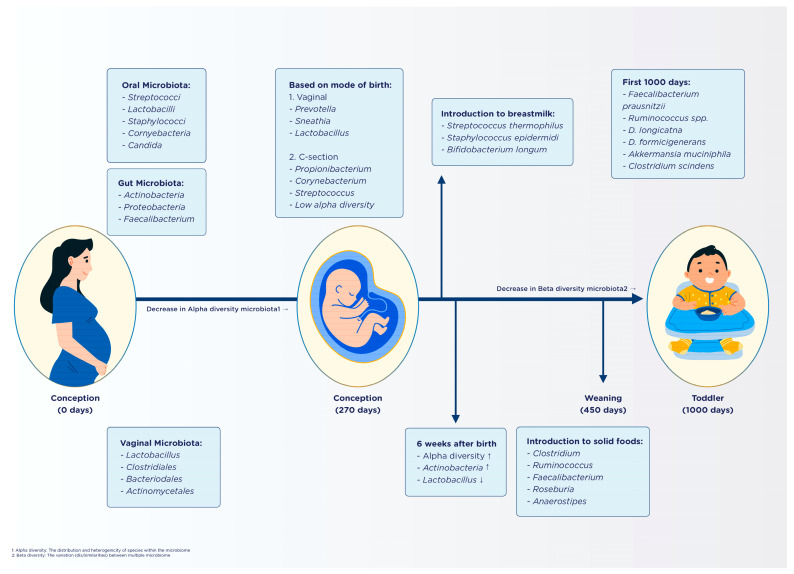
Temporal development of the gut microbiota during the first 1000 days of life [58,95,96].

**Figure 2 nutrients-15-02642-f002:**
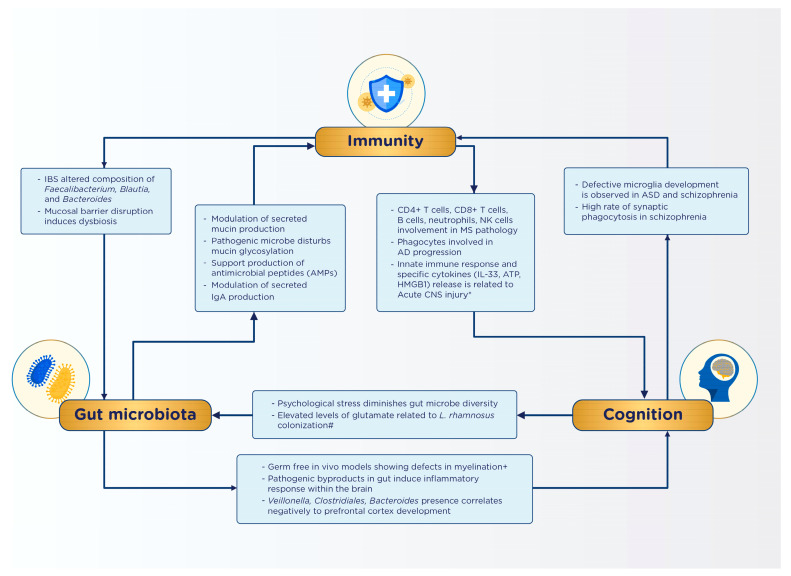
Summary of identified interactions in the gut–immunity–cognition axis [2,95,97,98,99,100,128].

**Table 1 nutrients-15-02642-t001:** Cytokines and their effect on neurodevelopment.

Interleukin	Function
IL-1 (IL-1A, IL-1B, IL-1RAP)	Neurogenesis, astrogliosis, neurite growth [110,111]
IL-4	Neurogenesis [112]
IL-33	Synaptic pruning, synaptic growth [113,114]
CSF1	Differentiation of immune cells [114]

## Data Availability

Not applicable.

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
