# Peer review of "Defining the Relationship of Gut Microbiota, Immunity, and Cognition in Early Life—A Narrative Review"

_nutrients, 2023, doi:10.3390/nu15122642_

Round 1
Reviewer 1 Report
The manuscript seems to me a narrative review, but the manuscript type is selected as an "article". The review does not meet all requirements of systematic review and they did not conduct a meta-analysis. Importantly results section contains many other refs than their 10 studies included for their analysis. Therefore I recommend to convert it to a proper narrative review.
English language and styles look ok. Maybe it requires a good proof reading after conversion.
Reviewer 2 Report
This is a manuscript on a current hot topic of gut microbiota-immune system-brain axis which could have been interesting. However, at present the manuscript has several limitations:
1. The manuscript is poorly written and cluttered. Extensive English editing is required. Authors should avoid describing well-established concepts which are common knowledge. Priority should be given to the evidences from published studies which support the central ideas. Unnecessary definitions should be avoided. Spellings and text duplication at many places needs to be corrected.
2. A more recent review is a very good read and relevant to this manuscript. Authors should try to incorporate this study in their references [Sittipo P, Choi J, Lee S, Lee YK. The function of gut microbiota in immune-related neurological disorders: a review. J Neuroinflammation. 2022 Jun 15;19(1):154. doi: 10.1186/s12974-022-02510-1. PMID: 35706008; PMCID: PMC9199126.]
3. All the figures except Figure 1 are of very poor resolution and unintelligible. Moreover, they are all labeled Figure 2. The manuscript needs to be revised fully for these errors before being considered further.
This is a manuscript on a current hot topic of gut microbiota-immune system-brain axis which could have been interesting. However, at present the manuscript has several limitations:
1. The manuscript is poorly written and cluttered. Extensive English editing is required. Authors should avoid describing well-established concepts which are common knowledge. Priority should be given to the evidences from published studies which support the central ideas. Unnecessary definitions should be avoided. Spellings and text duplication at many places needs to be corrected.
2. A more recent review is a very good read and relevant to this manuscript. Authors should try to incorporate this study in their references [Sittipo P, Choi J, Lee S, Lee YK. The function of gut microbiota in immune-related neurological disorders: a review. J Neuroinflammation. 2022 Jun 15;19(1):154. doi: 10.1186/s12974-022-02510-1. PMID: 35706008; PMCID: PMC9199126.]
3. All the figures except Figure 1 are of very poor resolution and unintelligible. Moreover, they are all labeled Figure 2. The manuscript needs to be revised fully for these errors before being considered further.
Reviewer 3 Report
The present review provides an extensive analysis of the relationship of gut-brain axis, immunity, and its influence on early life. Though the review is mostly well-written, however, there are some points that can be included or improvised. Below are my comments/suggestions:
1. There is excessive stress on methods used to find literature for the review and the same information is re-iterated. My recommendation is to trim down the section as much as possible and delete Figure 1.
2. If found repeat of information at many places. Please improve the redundancy of information in the review.
3. Apart from C-sections, please provide a comment on other early life stressors are known to affect the gut-brain axis and microbiota.
4. Figure 2 text is not legible.
5. Please comment on dietary interventions such as Probiotics, PUFAs to improve the impact of early-life attacks on gut microbiota.
6. Please provide more information of later life complications such as obesity, diabetes, eczema, and allergies that occur due to the disruption of gut microbiota due to early life stressors.
The quality of the language is fine.
Round 2
Reviewer 1 Report
Authors have improved the manuscript.
Reviewer 2 Report
Manuscript is now satisfactorily improved after incorporation of all the reviewers comments. The issues with the poor resolution of the figures remains as those are still not readable.
Manuscript is now satisfactorily improved after incorporation of all the reviewers comments. The issues with the poor resolution of the figures remains as those are still not readable.